# Resveratrol Reverses Endothelial Colony-Forming Cell Dysfunction in Adulthood in a Rat Model of Intrauterine Growth Restriction

**DOI:** 10.3390/ijms24119747

**Published:** 2023-06-05

**Authors:** Estelle Guillot, Anna Lemay, Manon Allouche, Sara Vitorino Silva, Hanna Coppola, Florence Sabatier, Françoise Dignat-George, Alexandre Sarre, Anne-Christine Peyter, Stéphanie Simoncini, Catherine Yzydorczyk

**Affiliations:** 1DOHaD Laboratory, Division of pediatrics, Department Woman-Mother-Child, Lausanne University Hospital, University of Lausanne, 1011 Lausanne, Switzerland; estelle.guillot@unil.ch (E.G.); anna.lemay@unil.ch (A.L.); manon.allouche@unil.ch (M.A.); sara.vitorinosilva@unil.ch (S.V.S.); hanna.coppola@unil.ch (H.C.); alexandre.sarre@chuv.ch (A.S.); 2Center from Cardiovascular and Nutrition Research (C2VN), Institut National de la Santé Et de la Recherche Médicale (INSERM), Aix Marseille Université, UMR-S 1263, UFR de Pharmacie, Campus Santé, 13385 Marseille, France; florence.sabatier-malaterre@univ-amu.fr (F.S.); francoise.dignat-george@univ-amu.fr (F.D.-G.); stephanie.simoncini@univ-amu.fr (S.S.); 3Institut National de Recherche pour L’Agriculture, L’Alimentation et L’Environnement (INRAe), Aix Marseille Université, UMR-S 1263, UFR de Pharmacie, Campus Santé, 13385 Marseille, France; 4Neonatal Research Laboratory, Clinic of Neonatology, Department Woman-Mother-Child, Lausanne University Hospital, University of Lausanne, 1011 Lausanne, Switzerland; anne-christine.peyter@chuv.ch

**Keywords:** intrauterine growth restriction, developmental programming, endothelial colony-forming cells, oxidative stress, stress-induced premature senescence, resveratrol

## Abstract

Individuals born after intrauterine growth restriction (IUGR) are at risk of developing cardiovascular diseases (CVDs). Endothelial dysfunction plays a role in the pathogenesis of CVDs; and endothelial colony-forming cells (ECFCs) have been identified as key factors in endothelial repair. In a rat model of IUGR induced by a maternal low-protein diet, we observed an altered functionality of ECFCs in 6-month-old males, which was associated with arterial hypertension related to oxidative stress and stress-induced premature senescence (SIPS). Resveratrol (R), a polyphenol compound, was found to improve cardiovascular function. In this study, we investigated whether resveratrol could reverse ECFC dysfunctions in the IUGR group. ECFCs were isolated from IUGR and control (CTRL) males and were treated with R (1 μM) or dimethylsulfoxide (DMSO) for 48 h. In the IUGR-ECFCs, R increased proliferation (5′-bromo-2′-deoxyuridine (BrdU) incorporation, *p* < 0.001) and improved capillary-like outgrowth sprout formation (in Matrigel), nitric oxide (NO) production (fluorescent dye, *p* < 0.01), and endothelial nitric oxide synthase (eNOS) expression (immunofluorescence, *p* < 0.001). In addition, R decreased oxidative stress with reduced superoxide anion production (fluorescent dye, *p* < 0.001); increased Cu/Zn superoxide dismutase expression (Western blot, *p* < 0.05); and reversed SIPS with decreased beta-galactosidase activity (*p* < 0.001), and decreased p16^ink4a^ (*p* < 0.05) and increased Sirtuin-1 (*p* < 0.05) expressions (Western blot). No effects of R were observed in the CTRL-ECFCs. These results suggest that R reverses long-term ECFC dysfunctions related to IUGR.

## 1. Introduction

Intrauterine growth restriction (IUGR), defined as a failure of the fetus to reach its full genetically determined growth potential, affects about 8–10% of all pregnancies and represents the second most common cause of perinatal mortality after prematurity. IUGR is characterized by an abnormal fetal growth slowing, resulting in a small body weight at birth compared with newborns with harmonious fetal growth at the same gestational age [1]. IUGR produces both short- and long-term consequences, and IUGR is associated with major perinatal mortality and morbidity, as well as with an increased risk of developing metabolic disorders and cardiovascular diseases (CVDs) [2]. Endothelial dysfunction plays a major role in the pathogenesis of cardiometabolic dysfunctions [3,4], notably in individuals born after IUGR [5]. Endothelial progenitor cells (EPCs) are critical circulating components of the endothelium and are identified as key factors in endothelial repair. Particularly, endothelial colony-forming cells (ECFCs) or late-outgrowth EPCs have clonal potential and the capacity to produce mature endothelial cells. They can proliferate, auto-renew, migrate, differentiate, and promote vascular growth and neovascularization [6].

Impaired number and functionality of ECFCs have been associated with several cardiometabolic disorders such as arterial hypertension and CVDs, obesity, type 2 diabetes, dyslipidemia, and nonalcoholic fatty liver diseases (NAFLD) [3]. In low birth weight (LBW) newborns and in a rat model of IUGR with cardiometabolic disorders at adulthood [7,8,9], the ECFCs isolated from, respectively, the cord blood and bone marrow, had an impairment in proliferation, vascular network formation, and angiogenic capabilities associated with oxidative stress and stress-induced premature senescence (SIPS) [10,11].

Resveratrol (3,5,40-trihydroxystilbene) (R) is a non-flavonoid polyphenolic compound that is a stilbene derivative. It is produced by plants and is present in grapes and red wine. Resveratrol has a protective effect against CVDs and could be involved in “the French paradox” where, despite a high intake of saturated fats, the incidence of CVDs is lower in the French population in association with moderate red wine consumption [12]. Among its protective effects against CVDs, it has been shown that resveratrol exerts an antihypertensive effect in several animal models of arterial hypertension such as the spontaneously hypertensive (SHR) rat, angiotensin (Ang) II-infused mouse, two-kidney one-clip hypertensive rat, and partially nephrectomized rats [13]. In addition, resveratrol could garner therapeutic interest in the management of blood pressure in the prehypertension and type 1 hypertension stages in patients [14]. Moreover, resveratrol has protective effects against atherosclerosis, stroke, myocardial ischemia, and heart failure [15] and improved flow-mediated vasodilation in several animal models [15,16,17]. A multitude of mechanisms have been identified to mediate these protective effects of resveratrol against CVDs, including antioxidant properties, the stimulation of endothelial nitride oxide synthase (eNOS) and nitric oxide (NO) production, the inhibition of vascular inflammation, the prevention of platelet aggregation [18,19,20,21,22], as well as through decreasing plasma triglycerides and low-density lipoprotein (LDL) cholesterol levels and increasing high-density lipoprotein (HDL) cholesterol [13].

In addition to its protective effects against CVDs, resveratrol also plays an important role in reproduction. In females, resveratrol is considered a natural phytoestrogen since its chemical structure is similar to that of some estrogens, such as diethylstilbestrol. Although some publications have shown a negative influence of resveratrol on female reproductive processes [23], several studies have demonstrated beneficial effects of resveratrol. It has been shown to promote ovarian cell culture, in vitro fertilization of oocytes, ovarian stimulation, ovarian follicular growth and oocyte maturation, embryonic development, cryopreservation of oocytes and embryos, and it can improve the quality of oocytes obtained from aged females [24,25,26]. These positive actions have been mainly associated with the antioxidant properties of resveratrol which decreases reactive oxygen species (ROS) production and improves antioxidant levels via an up-regulation of Sirtuin-1 [27,28] and decrease in the mammalian target of rapamycin (mTOR) expression [29]. In males, the role of resveratrol in reproductive function has not been clearly established. While Svechnikov et al. observed an inhibitory effect of resveratrol on steroidogenesis in the Leydig cells of rats [30], several studies subsequently showed that resveratrol can improve sperm quality in men by reducing germ cell apoptosis, improving penile erection, sperm quality, and sperm count in the epididymis, and increasing serum testosterone concentration [31,32]. Among the mechanisms that have been identified, resveratrol activates the AMP-activated protein kinase pathway, which improves mitochondrial function and the activity of antioxidant enzymes, therefore reducing ROS production [25].

In individuals born after IUGR, it has been shown that postnatal resveratrol supplementation improves metabolic [33] and ex vivo cardiac functions after ischemia/reperfusion injury in male and female IUGR offspring rats [34]. In ECFCs isolated from LBW newborns, Vassalo et al. demonstrated that resveratrol supplementation in vitro improved ECFC functionality associated with reversal of SIPS [10]. However, in these individuals, the supplementation with resveratrol was performed very early in postnatal life, which is considered a critical period of vulnerability; hence, whether resveratrol can exert a beneficial effect on the long-term dysfunctions related to IUGR has not been clearly identified. In this study, we proposed to explore, in a rat model of IUGR induced by altered maternal nutrition during gestation, the effects of resveratrol in vitro on the ECFC dysfunctions during adulthood (6 months of age) that we previously observed only in male IUGR offspring [11] (no differences were found in 6-month-old females). For this, we studied, in ECFCs isolated from bone marrow, the effect of resveratrol on ECFC proliferation and capillary-like structure formation capacities, NO production, oxidative stress response, and SIPS.

## 2. Results

### 2.1. Resveratrol Increased the Proliferation Capacity in IUGR-ECFCs

We explored the effect of resveratrol on the proliferative capacity of ECFCs from the control (CTRL) and IUGR groups by measuring the absorbance at 450 nm at 6 and 24 h after 5′-bromo-2′-deoxyuridine (BrdU) incorporation. In the IUGR-ECFCs, resveratrol significantly increased cell proliferation at 6 and 24 h (+107% at 6 h; +238% at 24 h), whereas no difference was observed in the CTRL ± resveratrol groups (Figure 1).

### 2.2. Resveratrol Improved the Capillary-like Structure Formation in IUGR-ECFCs

Then, we evaluated the effect of resveratrol on the capillary-like structure formation capacity of the ECFCs from the CTRL and IUGR groups. Resveratrol treatment increased the endothelial network formation by the IUGR-ECFCs, as shown by an increased number of closed tubes and branches compared with the CTRL group. We observed no difference in the CTRL ± resveratrol groups (Figure 2).

### 2.3. Resveratrol Improved the NO Production and eNOS Expression in IUGR-ECFCs

The NO production of the ECFCs ± resveratrol was assessed using the fluorescent dye 4,5-diaminofluorescein diacetate (DAF-2DA). In the IUGR-ECFCs, the resveratrol treatment significantly increased NO production under basal conditions (+51%; *p* < 0.01), as well as after stimulation with acetylcholine (+64%; *p* < 0.001) (Figure 3 and Figure 4). No difference was observed in the CTRL ± resveratrol groups (Figure 3 and Figure 4). In addition, resveratrol increased eNOS expression in the IUGR-ECFCs as measured by immunofluorescence (+56%; *p* < 0.001) (Figure 5). No difference was found in the CTRL ± resveratrol groups (Figure 5).

### 2.4. Resveratrol Decreased Oxidative Stress in IUGR-ECFCs

Superoxide anion production was assessed using the oxidative fluorescent dye hydroethidine. A significant decrease in superoxide anion production (−63%; *p* < 0.001) was observed in the resveratrol-treated IUGR-ECFCs (Figure 6A). In addition, resveratrol improved the Cu/Zn superoxide dismutase (SOD) protein expression (+85%; *p* < 0.05) (Figure 6B). No effect of resveratrol was observed concerning the superoxide anion production and the expression of Cu/Zn SOD in the CTRL-ECFCs (Figure 6A,B).

### 2.5. Resveratrol Reversed Stress-Induced Premature Senescence in IUGR-ECFCs

The effect of treatment with resveratrol on the cellular senescence was investigated by measuring senescence-associated beta-galactosidase (SA-β-gal) activity. In the IUGR-ECFCs, resveratrol significantly reduced the percentage of senescent cells (−37%; *p* < 0.001) (Figure 7).

We measured the expression of two senescence markers: Sirtuin-1 and the cyclin-dependent kinase (CDK) inhibitor p16^INK4a^. Resveratrol treatment of the IUGR-ECFCs increased Sirtuin-1 protein expression according to the Western blot and immunofluorescence results (+24%; *p* < 0.05 and +52%; *p* < 0.001, respectively) (Figure 8 and Figure 9) and decreased p16^INK4a^ protein expression (−43%; *p* < 0.05) (Figure 9). In the CTRL-ECFCs, no effect of resveratrol was observed concerning the SA-β-gal activity, or Sirtuin-1 and p16^INK4a^ protein expression (Figure 7, Figure 8 and Figure 9).

## 3. Discussion

The beneficial effects of resveratrol treatment on long-term ECFC dysfunctions related to IUGR have not been clearly identified. The findings from this study demonstrate for the first time that in vitro treatment with 1 μM resveratrol had beneficial effects by reversing the ECFC dysfunctions observed in IUGR male rats at 6 months of age [11]. Resveratrol improved the proliferation and formation of capillary-like structures, increased NO production and eNOS expression, decreased oxidative stress, and reversed SIPS.

A previous study from our research group showed altered ECFC function in adulthood in a rat model of IUGR. Indeed, we observed that at six months of age, the IUGR-ECFCs had a reduced capacity to proliferate and impaired capillary-like structure formation, reflecting a lack of capacity to repair vascular damage [11]. In this study, we observed that, in the IUGR-ECFCs, 1 μM resveratrol improved the proliferation as well as the capillary-like structure formation with increased closed tubes and branches formation. Similar findings were observed in the ECFCs isolated from LBW newborns [10]. Our observations are also consistent with the capacity of resveratrol to improve endothelial progenitor function in vitro [10,35,36,37,38].

The proper functioning of ECFCs and angiogenesis are strongly related. Our previous work showed that the impaired function of IUGR-ECFCs was associated with altered angiogenesis, and decreased NO production and eNOS expression [11]. Depending on the situation, resveratrol can exert pro- and anti-angiogenic effects. Indeed, pro-angiogenic effects have been noted in the peri-infarct myocardium, whereas anti-angiogenic effects have been observed principally in tumors [39]. The pro-angiogenic effect of resveratrol has been associated with the up-regulation of NO production [15,40]. In this study, we observed that 1 μM resveratrol treatment improved NO production by increasing eNOS expression in the IUGR-ECFCs, consistent with results observed in endothelial cells [41]. NO, which is necessary for angiogenesis to occur [42], is involved in the mobilization of EPCs and improves their migratory and proliferative activities [43]. Therefore, increased NO production could explain the beneficial effects of resveratrol on the proliferation and capillary-like structure formation observed in the IUGR-ECFCs. Our observations were consistent with other studies describing resveratrol as a potential modulator of endothelial functions in several pathologies associated with IUGR such as preeclampsia or maternal hyperglycemia [40]. In addition, resveratrol has been shown to enhance NO production notably by preventing eNOS uncoupling by reducing oxidative stress [21,40,44].

Our previous work showed increased superoxide anion production associated with decreased Cu/Zn SOD in IUGR-ECFCs [11]. In vitro, resveratrol has been reported to directly scavenge a variety of ROS, especially superoxide anions, at high concentrations (≥100 μM) [45]. Interestingly, in this study, we observed that at a low concentration (1 μM) of resveratrol was able to decrease the superoxide anion level in the IUGR-ECFCs. The data from cell cultures and animal models have shown that resveratrol (10–100 μM) can increase the SOD mRNA and protein levels [46,47,48]. In the IUGR-ECFCs, we observed that 1 μM resveratrol was able to increase expression of the antioxidant Cu/Zn SOD, which catalyzes the dismutation of superoxide anion to hydrogen peroxide (H_2_O_2_), which could explain the decreased superoxide anion level mentioned above. Among the key molecules involved in the regulation of oxidative stress, Sirtuin-1 has been shown to play an important role [49,50]. Our previous work observed decreased Sirtuin-1 protein expression in IUGR-ECFCs [11]. In this study, resveratrol increased Sirtuin-1 expression in the IUGR-ECFCs which could also be associated with a decreased superoxide anion concentration due to the activation of antioxidant defense mechanisms [48]. Sirtuin-1 also plays an important role in cellular senescence [51]. Sirtuin-1 can decrease the levels of the CDK inhibitor p16^INK4a^, preventing cellular senescence [10,52]. In addition, it has been demonstrated that Sirtuin-1 is activated by calorie restriction [53] and binds directly to the p16^INK4a^ promoter, decreasing its expression through deacylation by Sirtuin-1, which contributes to delaying senescence/aging processes [52]. Resveratrol has been identified as a potent activator of Sirtuin-1 [54] and a mimic of calorie restriction [50]; therefore, resveratrol also plays an important role in cellular senescence. It has been shown that high concentrations of resveratrol (≥50 μM) can induce premature senescence [55]; however, our research group previously demonstrated that at a low concentration (1 μM), resveratrol can reverse SIPS in ECFCs isolated from cord blood in LBW newborns in a Sirtuin-1-dependent manner [10]. Our previous study demonstrated that in IUGR-ECFCs, in addition to decreased Sirtuin-1 protein expression, there was an increase in SA-β-gal activity, which has been identified as the gold standard for senescence detection [56,57], and p16^INK4a^ protein expression, reflecting the presence of SIPS [11]. In this study, we observed that 1 μM resveratrol treatment decreased SA-β-gal activity in the IUGR-ECFCs, which was also observed in animal and cell line models of senescence/aging [58,59]. In addition, 1 μM resveratrol decreased p16^INK4a^ protein expression, as shown in the ECFCs isolated from LBW newborns [10]. These data suggest that 1 μM resveratrol reversed the SIPS observed in the IUGR-ECFCs, which could be due to the up-regulation of Sirtuin-1 expression.

## 4. Materials and Methods

### 4.1. Animal Model

The animal model used in this study was described previously [9,11,60]. Briefly, a rat model of IUGR was induced by an altered maternal diet during gestation. Pregnant rats (Sprague Dawley) were randomly allocated to a control diet (20% casein; CTRL group) or to an isocaloric low-protein (LP) diet (8% casein; IUGR group) throughout gestation. The male and female pups in the IUGR group displayed LBWs compared with males and females in the CTRL group, and the difference persisted into adulthood [11]. At 6 months of age, the animals were euthanized by an intraperitoneal injection of pentobarbital (Esconarkon, Streuli Pharma AG, Uznach, Switzerland) at a dose of 150 mg/kg of body weight followed by exsanguination. In this study, only ECFCs isolated from male rats were studied.

### 4.2. Endothelial Progenitor Cell Isolation and Treatment

Bone marrow was collected from the tibialis and femur of the CTRL and IUGR male rats at six months of age, as previously described [11]. After density gradient centrifugation (Histopaque 1077-, Sigma-Aldrich, Saint Louis, MO, USA), the mononuclear cells isolated from the interface were washed in Roswell Park Memorial Institute (RPMI) medium with 10% fetal calf serum (Thermo Fisher Scientific, Rockford, IL, USA), and resuspended in endothelial basal cell growth culture medium-2 (EBM2) supplemented with an endothelial cell growth medium MV2 (PromoCell, Heidelberg, Germany) and penicillin/streptomycin (Sigma-Aldrich). The colonies of ECFCs were identified as well-circumscribed monolayers of cobblestone-appearing cells using an inverted microscope (Eclipse Ti2 Series-Nikon Europe B.V., Amsterdam, The Netherlands) as previously described [7,11]. The cells were isolated from the CTRL (CTRL-ECFCs) and IUGR (IUGR-ECFCs) male rats and were studied between passages 1 and 3. The ECFC experiments represent individual animals taken from separate litters. Unfortunately, primary cultures are particularly sensitive, and during the experiments, there was some contamination, and we had to discard some ECFCs, which explains the difference in numbers between the experiments.

Drug treatment

The ECFCs were incubated for 48 h with 1 µM R (Calbiochem, La Jolla, San Diego, CA, USA) in dimethylsulfoxide (DMSO; Sigma-Aldrich). The control cells were treated with DMSO [10].

### 4.3. ECFC Proliferation Test

The measurement of 3H thymidine incorporation as cells enter the S phase has long been the reference method for the detection of cell proliferation. However, this technique requires the quantification of 3H thymidine using scintillation counting or autoradiography. Therefore, an alternative approach has been developed in which BrdU, a thymidine analog, replaces the 3H thymidine. We measured the proliferative capacity of the CTRL ± R (*n* = 4) and IUGR ± R (*n* = 5) ECFCs (20,000 cells/well) after 6 and 24 h using a colorimetric cell proliferation ELISA test based on the incorporation of BrdU into newly synthesized DNA strands of actively proliferating cells (Roche diagnostics, Basel, Switzerland). Each experiment was performed in duplicate [11].

### 4.4. The Capillary-like Structure Formation

The development of capillary-like structure in ECFCs was assessed using Matrigel™ BD Growth factor (Becton, Dickinson and Company, Franklin Lakes, NJ, USA). Aliquots of Matrigel were prepared on ice under sterile conditions. Then, we proceeded to coat the 96-well plates with Matrigel (50 μL per well). The CTRL ± R (*n* = 4) and IUGR ± R (*n* = 5) ECFCs were seeded at a concentration of 20,000 cells/well into the 96-well plates coated with Matrigel. Each experiment was performed in duplicate. After 24 h, pictures were blindly obtained using an inverted microscope (Eclipse Ti2 Series) by the same examiner (E.G.) [11].

### 4.5. Measurement of NO Production by ECFCs

DAF-2 DA is a membrane-permeable fluorescent dye, which allows us to measure NO production from living cells. The CTRL ± R (*n* = 4) and IUGR ± R (*n* = 5) ECFCs were incubated with DAF-2DA (10 mM; Merck Millipore, Darmstadt, Germany) in a light-protected, humidified chamber at 37 °C for 1 h and then incubated for 1 h at 37 °C in N-2-Hydroxyethylpiperazine-N-2-Ethane Sulfonic Acid (HEPES) buffer alone or with acetylcholine (100 mM) added (Sigma-Aldrich). The ECFCs were rinsed in HEPES and then mounted on slides using fluoromount-G mounting medium with DAPI (Life Technologies Europe B.V, Zug, Switzerland). A negative control was established through incubation without DAF-2DA. The fluorescence of DAF-2DA was normalized to the DAPI fluorescence and the autofluorescence was subtracted. Each experiment was performed in duplicate. The images at ×20 magnification were blindly obtained using an inverted fluorescent microscope (Eclipse Ti2 Series) and analyzed with ImageJ software (Java 1.8.0_112, National Institutes of Health, Montgomery, AL, USA, accessed on 1 July 2021) [11].

### 4.6. Measurement of Superoxide Anion Production by ECFCs

Superoxide anion production was measured using the oxidative fluorescent dye hydroethidine. In the presence of superoxide anions, hydroethidine is oxidized to fluorescent ethidium bromide, which is trapped in DNA by intercalation. The superoxide anion production of the CTRL ± R (*n* = 4) and IUGR ± R (*n* = 5) ECFCs was evaluated by incubating them with hydroethidine (2 μM; Sigma-Aldrich) and then mounting them on slides with fluoromount-G mounting medium-DAPI (Life Technologies). A negative control was established through incubation without hydroethidine. The fluorescence of superoxide anions was normalized to the DAPI fluorescence, and the autofluorescence was subtracted. Each experiment was performed in duplicate. The images at ×20 magnification were blindly obtained using an inverted fluorescent microscope (Eclipse Ti2 Series) and analyzed with ImageJ software (Java 1.8.0_112) by the same examiner (E.G.) [11].

### 4.7. Senescence Detection in ECFCs

The SA-β-gal is an enzyme which hydrolyzes the glycosylic bonds in X-Gal, producing an easily recognizable blue reaction product. Its optimum activity is reached at pH 6.0, which is a characteristic of senescent cells. The SA-β-gal activity was evaluated in the CTRL ± R (*n* = 4) and IUGR ± R (*n* = 4) ECFCs using a senescence detection kit (Cell Signaling Technology, Danvers, MA, USA) according to the manufacturer’s instructions. The SA-β-gal-positive cells (blue staining) were normalized as a percentage of the total number of cells. Each experiment was performed in duplicate. Using an inverted microscope (Eclipse Ti2 Series), the pictures at ×20 magnification were blindly acquired and analyzed with ImageJ software (Java 1.8.0_112) by the same examiner (E.G.) [10,11].

### 4.8. Protein Expression Measurement Using Western Blotting

Proteins from the CTRL ± R (*n* = 3–4) and IUGR ± R (*n* = 5) ECFCs were extracted at 6 months of age, as previously described [11]. Denatured (10 min at 70 °C) proteins (35 μg) were separated on a gradient gel (NuPAGE 4–12% Bis-Tris gel; Life Technologies) and transferred overnight at 4 °C to Whatman nitrocellulose membranes (Life Technologies). Ponceau staining (Life Technologies) confirmed the presence of proteins on the membrane. Primary antibodies against Cu/Zn SOD, catalase, Sirtuin-1, p16^INK4a^, and β-actin were purchased (Cell Signaling and Abcam (Cambridge, UK)) and used at the dilutions recommended for immunoblotting (1:1000). The antibody dilutions were performed in the blocking buffer (tris-buffered saline (TBS; Sigma-Aldrich)-tween (Sigma-Aldrich) 1%-bovine serum albumin (BSA 3%; AppliChem GmbH, Darmstadt, Germany) and incubated overnight at 4 °C. The incubations with anti-mouse or anti-rabbit secondary antibodies (1/2000; Cell Signaling) were performed for 2 h at room temperature in TBS-tween 1%-BSA 3%. The antibodies were visualized using an enhanced chemiluminescence Western blotting substrate (Life Technologies). A G-BOX Imaging System (GeneSys, Syngene, Cambridge, UK) was used to detect specific bands, and the optical density of each band was measured using specific software (GeneTools 4.03.05.0, Syngene) for all blots. For technical reasons, it was not possible to load the CTRL and IUGR on the same blot. For IUGR-ECFCs, we conducted the Western blotting for Sirtuin-1 and Cu/Zn SOD on the same membrane to reduce the biological material consumption.

### 4.9. Immunofluorescence

The CTRL ± R (*n* = 4) and IUGR ± R (*n* = 5) ECFCs were fixed using cold 70% ethanol (Life Technologies) and were incubated with antibodies against Sirtuin-1 and eNOS (rabbit, 1:100; cell signaling) overnight at 4 °C. The ECFCs were then washed with phosphate-buffered saline (PBS; Life Technologies), incubated for 2 h with Alexa Fluor-488 goat anti-rabbit IgG (IgG 1:200) (Abcam), rinsed with PBS, and then mounted on slides with fluoromount-G mounting medium-DAPI (Life Technologies). A negative control was obtained by only incubating with the secondary antibody. The fluorescence of Sirtuin-1 and eNOS was normalized to the DAPI fluorescence and the autofluorescence was subtracted. Each experiment was performed in duplicate. The images at ×20 magnification were blindly obtained using an inverted fluorescent microscope (Eclipse Ti2 Series) and analyzed with ImageJ software (Java 1.8.0_112) by the same examiner (E.G.) [11].

### 4.10. Statistical Analyses

All data are presented as the mean ± standard deviation (SD). After confirming a normal distribution, the experimental observations were analyzed using Student’s t-test. GraphPad Prism 9 (version 9.1.0 (221), La Jolla, CA, USA) was used for the statistical analyses and creating the graphics. The significance level was set at *p* < 0.05.

## 5. Conclusions

The present study is, to our knowledge, the first to demonstrate that an in vitro resveratrol treatment can reverse the ECFC dysfunctions at 6 months of age in male rats born after IUGR. In fact, resveratrol improved the proliferation and the capillary-like structure formation of ECFCs and increased their NO production and eNOS expression. These positive effects of resveratrol on IUGR-ECFC function were associated with decreased oxidative stress and reversal of SIPS (Figure 10).

Limitations

We had a limited number of ECFCs in this study because these cells were difficult to isolate, and primary cultures are particularly sensitive. During the experiments, there was some contamination, and we had to discard some ECFCs. Therefore, we chose to use only the dose and timing of resveratrol previously used by Vassalo et al. [10] because we did not have enough cells to perform experiments to evaluate the time- and dose-dependent effects.

Perspectives

To confirm that the effects of resveratrol on oxidative stress and SIPS were due to the up-regulation of Sirtuin-1, it would be of interest to over-express Sirtuin-1 in ECFCs using lentivirus infection and to treat the ECFCs with nicotinamide, an inhibitor of Sirtuin-1.

Finally, as resveratrol exerts positive effects on reproductive processes, it would be of interest to test in vivo whether an LP diet supplemented with resveratrol during gestation could prevent ECFC dysfunctions in the offspring and thereby delay/prevent the development of the cardiovascular and metabolic disorders that we previously observed in IUGR males during adulthood [9,11].

## Figures and Tables

**Figure 1 ijms-24-09747-f001:**
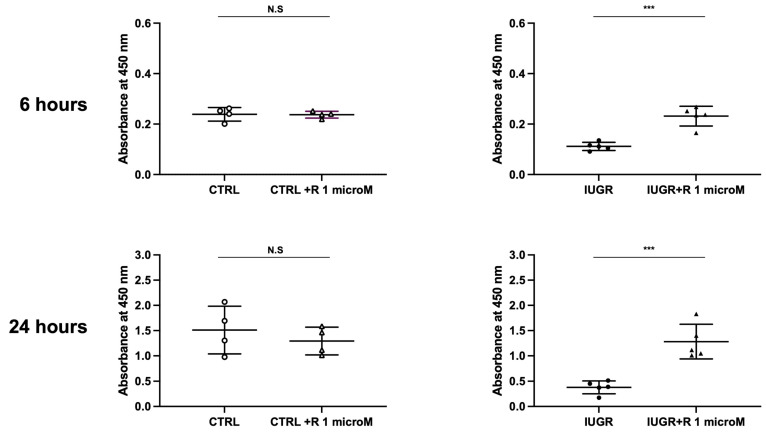
Proliferation capacities of the CTRL ± R and IUGR ± R ECFCs isolated from six-month-old male rats using BrdU incorporation at 6 and 24 h. = 4–5 animals/group; *** *p* < 0.001; N.S.: not significant. R: resveratrol.

**Figure 2 ijms-24-09747-f002:**
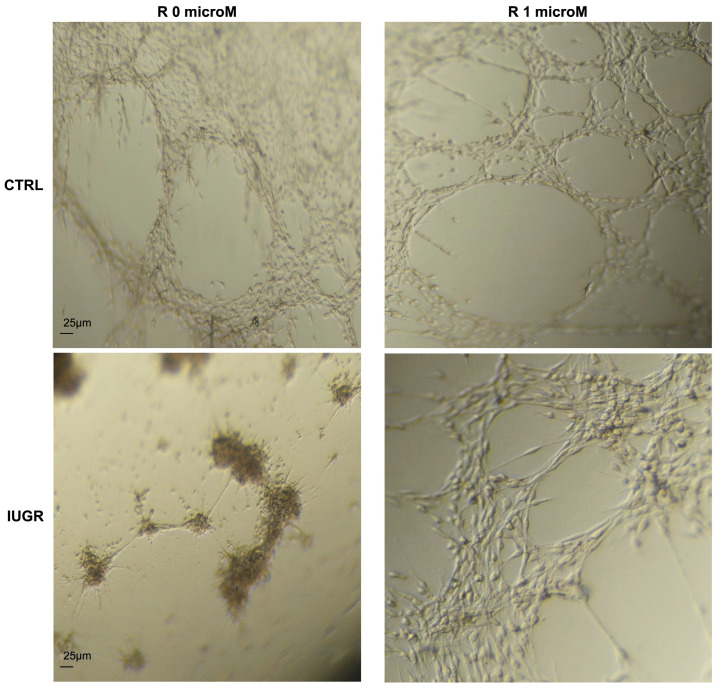
The capillary-like outgrowth sprouts were evaluated in 24 h Matrigel cultures of the CTRL ± R and IUGR ± R ECFCs isolated from six-month-old male rats. Magnification = 5×. These pictures are representative images from *n* = 4–5 animals/group. Scale bar = 25 μm. R: resveratrol.

**Figure 3 ijms-24-09747-f003:**
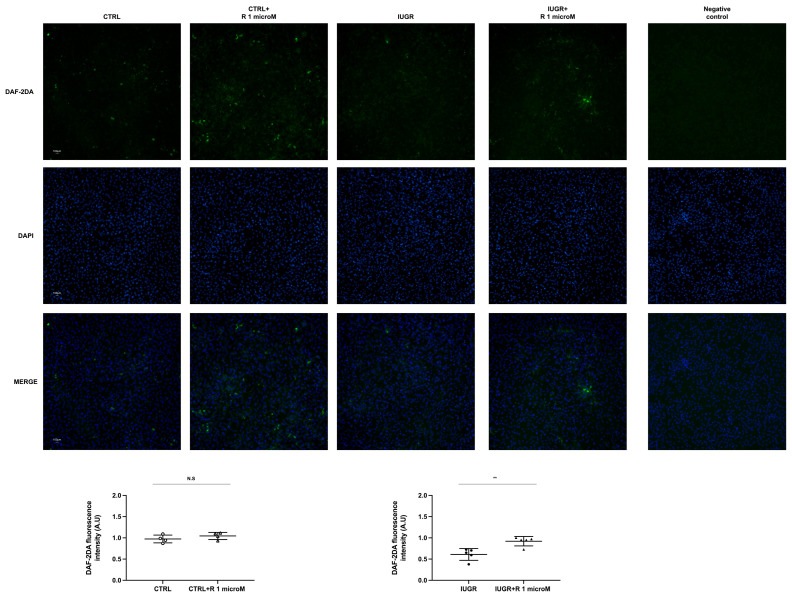
Basal NO production. Measurement of NO production using DAF-2DA was performed using the CTRL ± R and IUGR ± R ECFCs isolated from six-month-old male rats under baseline conditions. Magnification = 20×. Nuclei were counterstained with 4′6-diamidino-2-phenylindole (DAPI). A negative control was performed. These pictures are representative images from *n* = 4–5 animals/group; ** *p* < 0.01; N.S.: not significant. Scale bar = 100 μm. R: resveratrol.

**Figure 4 ijms-24-09747-f004:**
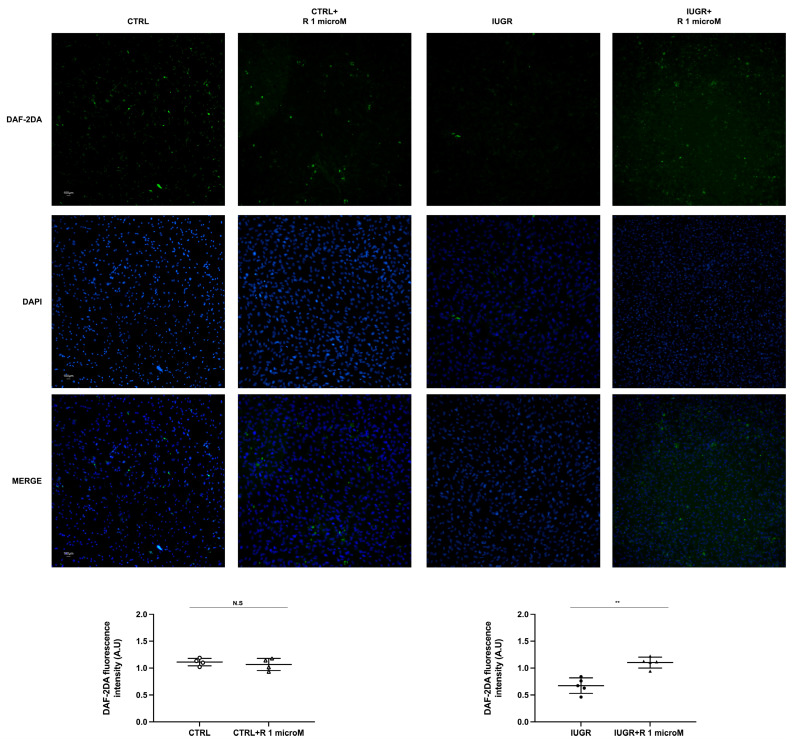
Stimulated NO production using acetylcholine. Measurement of NO production using DAF-2DA was performed using the CTRL ± R and IUGR ± R ECFCs isolated from six-month-old male rats after stimulation with acetylcholine (100 mM). Magnification = 20×. Nuclei were counterstained with DAPI. These pictures are representative images from *n* = 4–5 animals/group; ** *p* < 0.01; N.S.: not significant. Scale bar = 100 μm. R: resveratrol.

**Figure 5 ijms-24-09747-f005:**
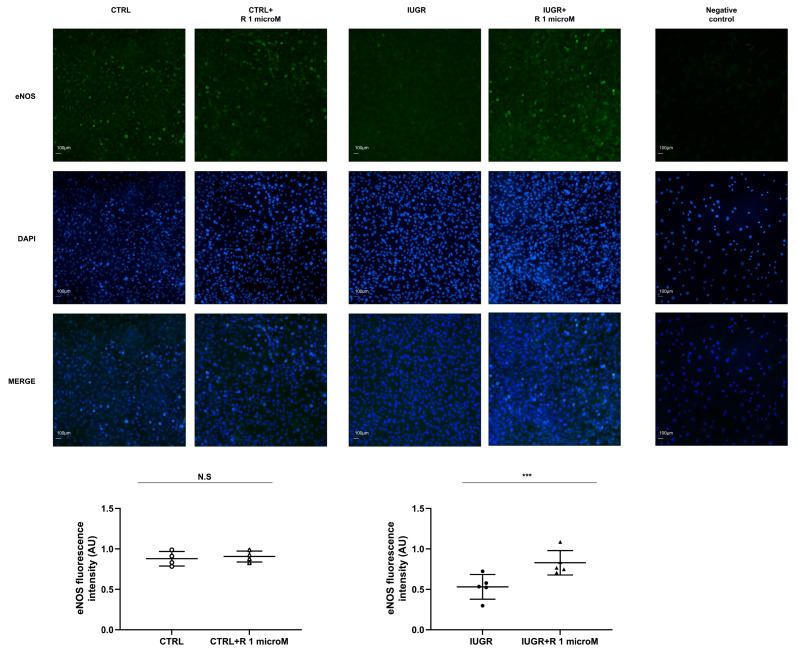
eNOS protein expression. eNOS protein expression was measured using immunofluorescence in the CTRL ± R and IUGR ± R ECFCs isolated from six-month-old male rats. Magnification = 20×. Nuclei were counterstained with DAPI, and a negative control was performed. These pictures are representative images from *n* = 4–5 animals/group; *** *p* < 0.001; N.S.: not significant. Scale bar = 100 μm. R: resveratrol.

**Figure 6 ijms-24-09747-f006:**
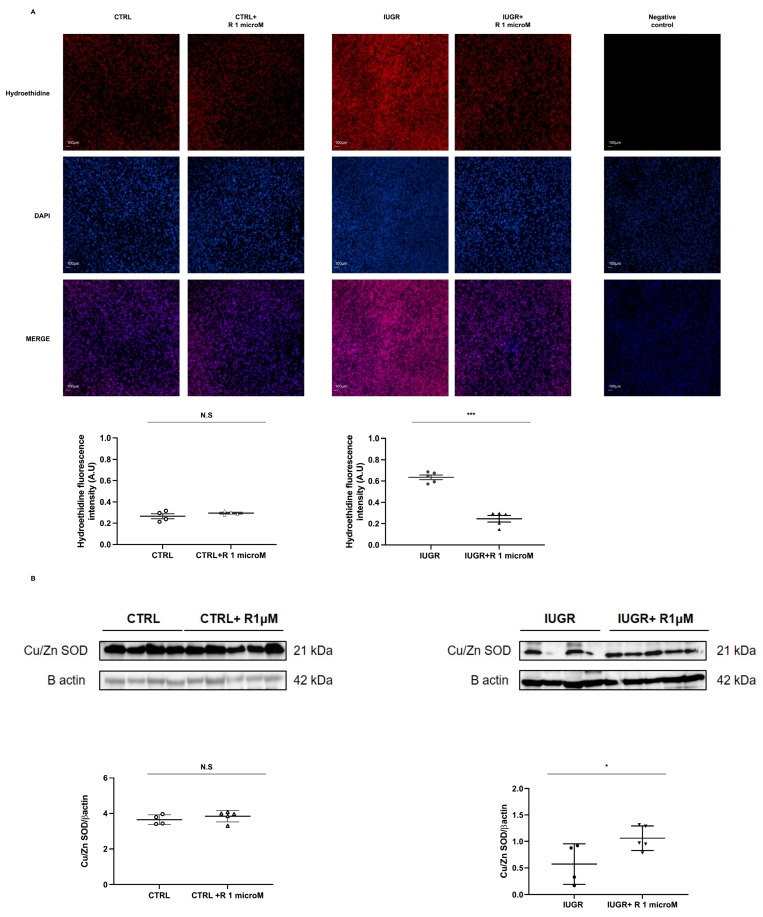
Evaluation of oxidative stress in the ECFCs. Superoxide anion production was measured using hydroethidine in the CTRL ± R and IUGR ± R ECFCs isolated from six-month-old male rats (**A**). Magnification = 20×. Nuclei were counterstained with DAPI and a test for autofluorescence was performed. These pictures are representative images from *n* = 4–5 animals/group. Scale bar = 100 μm. In addition, the antioxidant Cu/Zn SOD protein expression was evaluated using Western blotting of the CTRL ± R and IUGR ± R ECFCs isolated from six-month-old male rats (**B**). *n* = 4–5 animals/group; * *p* < 0.05; *** *p* < 0.001; N.S.: not significant. R: resveratrol.

**Figure 7 ijms-24-09747-f007:**
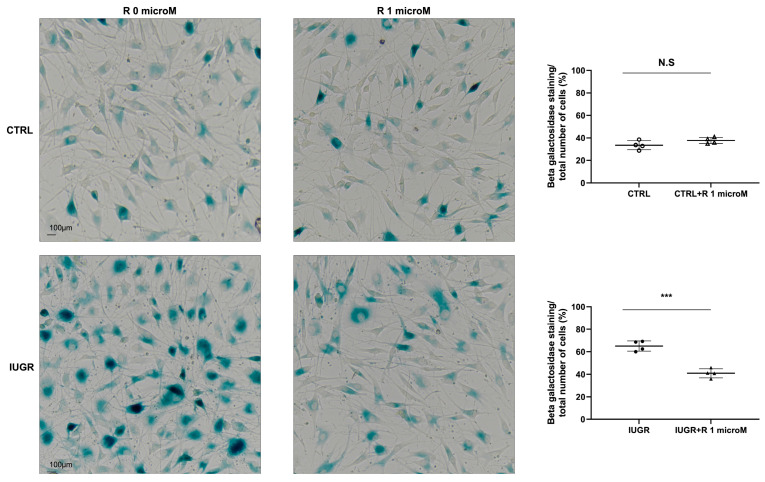
SA-β-gal activity in the ECFCs. The SA-β-gal activity determined as the blue staining normalized to the total number of cells in the CTRL ± R and IUGR ± R ECFCs isolated from six-month-old male rats. Magnification = 20×. These pictures are representative images from *n* = 4 animals/group; *** *p* < 0.001. Scale bar = 100 μm. N.S.: not significant. R: resveratrol.

**Figure 8 ijms-24-09747-f008:**
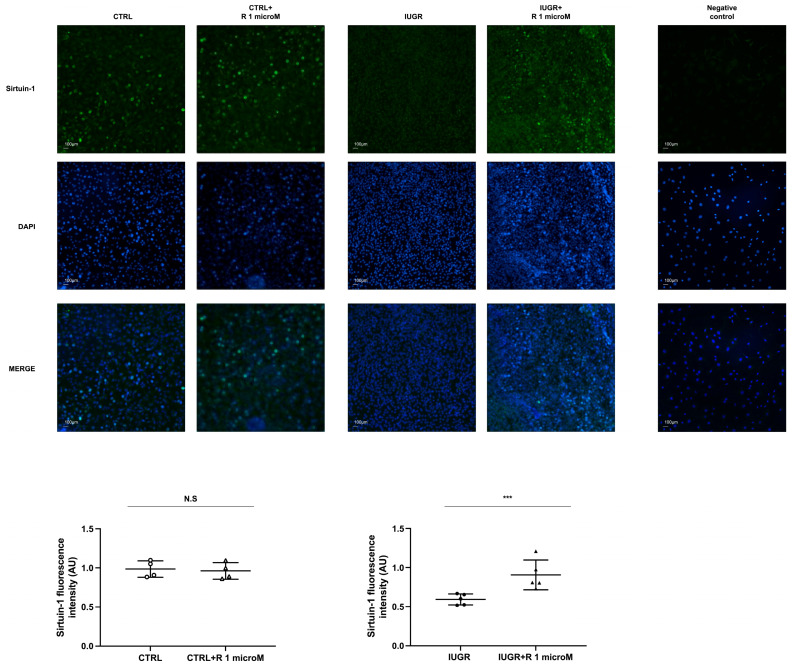
Sirtuin-1 expression. Sirtuin-1 expression was measured by immunofluorescence in the CTRL ± R and IUGR ± R ECFCs isolated from six-month-old male rats. Nuclei were counterstained with DAPI, and a negative control was performed. Magnification = 20×. These pictures are representative images from *n* = 4–5 animals/group; *** *p* < 0.001; N.S.: not significant. Scale bar = 100 μm. R: resveratrol.

**Figure 9 ijms-24-09747-f009:**
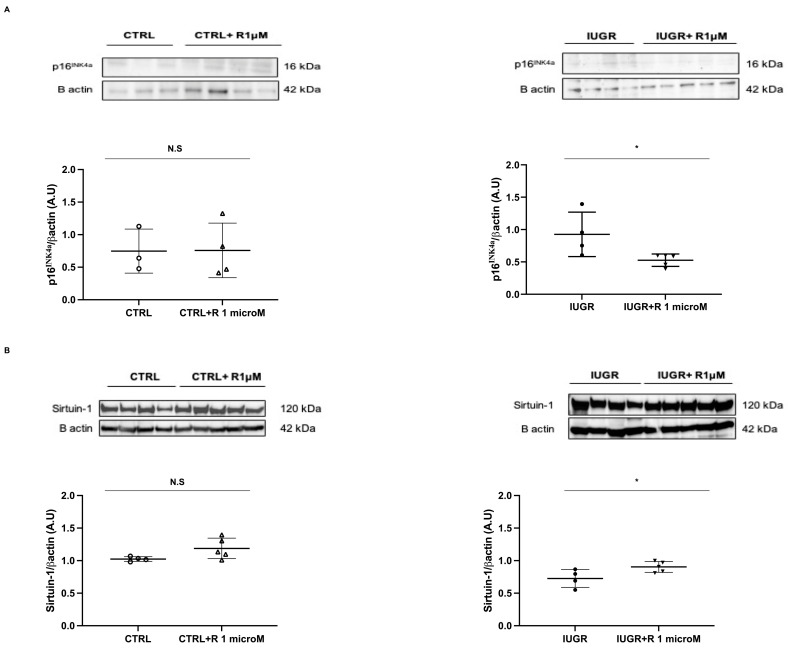
Factors related to cellular senescence. p16^INK4a^ (**A**) and Sirtuin-1 (**B**) protein expressions were measured using Western blotting in the CTRL ± R and IUGR ± R ECFCs isolated from six-month-old male rats. *n* = 3–5 animals/group; * *p* < 0.05; N.S.: not significant. R: resveratrol.

**Figure 10 ijms-24-09747-f010:**
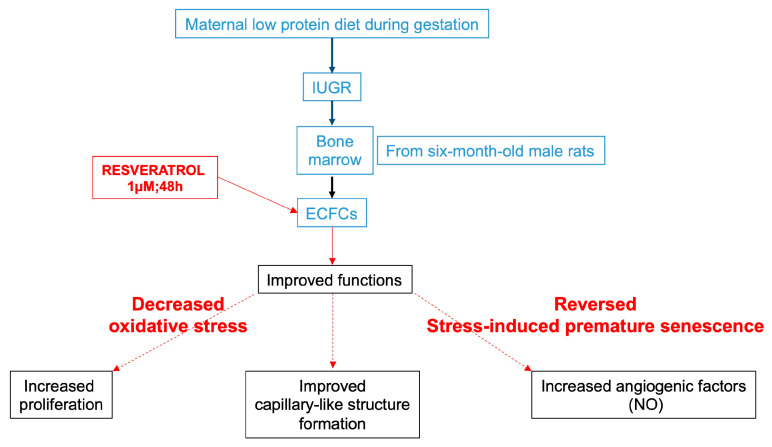
The effects of resveratrol on the ECFC dysfunctions related to IUGR at 6 months of age.

## Data Availability

Not applicable.

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
