# Peer review of "Resveratrol Reverses Endothelial Colony-Forming Cell Dysfunction in Adulthood in a Rat Model of Intrauterine Growth Restriction"

_ijms, 2023, doi:10.3390/ijms24119747_

Round 1

Reviewer 1 Report

Overall the manuscript is very interesting and contains very useful information for scientists of the same field. I have only these few suggestions.

L47-48 Please check the line alignment

L52 check comma

Resveratrol function introduction: It would be interesting to add, since the paper deal with a model of intrauterine growth restriction, some bibliographic references to the action of resveratrol in reproduction such as:

The Role of Resveratrol in Mammalian Reproduction. Molecules. 2020 Oct 5;25(19):4554. doi: 10.3390/molecules25194554. 

Reviewer 2 Report

I read with interest the article on the beneficial effects of resveratrol on endothelial cells.
In this study, Estelle Guillot et al showed that 1µM resveratrol on endothelial colony-forming cells improved proliferation and capillary-like structure, increased production of NO, decreased oxidative stress, and reversed stress-induced premature senescence. I have the following comments and suggestions:

There are some typos and editing errors in the manuscript that detract in a significant way from the readability of the manuscript. The manuscript should be carefully revised before it is intended for publication. For example, (1) Lines 47-48, 163-164. (2) Abbreviations (e.g., CTRL, IUGR) and their full form should be provided wherever they appear first time in the manuscript and later use abbreviation.

Literature must be upgraded. I want to underline that 43 references (out of 5) are older than 5 years.

There are some typos and editing errors in the manuscript that detract in a significant way from the readability of the manuscript. 

Reviewer 3 Report

Dear Editor

The manuscript is clear and methodologically accept.

I have no comments 
